# Immunogenicity and Immune Memory after a Pneumococcal Polysaccharide Vaccine Booster in a High-Risk Population Primed with 10-Valent or 13-Valent Pneumococcal Conjugate Vaccine: A Randomized Controlled Trial in Papua New Guinean Children

**DOI:** 10.3390/vaccines7010017

**Published:** 2019-02-04

**Authors:** Anita H. J. van den Biggelaar, William S. Pomat, Geraldine Masiria, Sandra Wana, Birunu Nivio, Jacinta Francis, Rebecca Ford, Megan Passey, Lea-Ann Kirkham, Peter Jacoby, Deborah Lehmann, Peter Richmond

**Affiliations:** 1Wesfarmers Centre of Vaccines and Infectious Diseases, Telethon Kids Institute, University of Western Australia, Nedlands, WA 6009, Australia; lea-ann.kirkham@uwa.edu.au (L.-A.K.); deborah.lehmann@telethonkids.org.au (D.L.); 2Division of Paediatrics, School of Medicine, University of Western Australia, Crawley, WA 6009, Australia; 3Papua New Guinea Institute of Medical Research, Goroka, Eastern Highlands Province, Papua New Guinea; william.pomat@pngimr.org.pg (W.S.P.); geraldine.masiria@pngimr.org.pg (G.M.); numbocquio@gmail.com (S.W.); birunu.nivio@pngimr.org.pg (B.N.); jacintafrancis@gmail.com (J.F.); rebecca.ford@pngimr.org.pg (R.F.); 4School of Public Health, University Centre for Rural Health (USRH), The University of Sydney, Lismore, NSW 2480, Australia; megan.passey@sydney.edu.au; 5School of Biomedical Sciences, University of Western Australia, Crawley, WA 6009, Australia; 6Centre for Biostatistics, Telethon Kids Institute, Nedlands, WA 6009, Australia; peter.jacoby@telethonkids.org.au

**Keywords:** pneumococcal polysaccharide vaccine, pneumococcal conjugate vaccine, *S. pneumoniae*, antibodies, immune memory, Papua New Guinea

## Abstract

We investigated the immunogenicity, seroprotection rates and persistence of immune memory in young children at high risk of pneumococcal disease in Papua New Guinea (PNG). Children were primed with 10-valent (PCV10) or 13-valent pneumococcal conjugate vaccines (PCV13) at 1, 2 and 3 months of age and randomized at 9 months to receive PPV (PCV10/PPV-vaccinated, n = 51; PCV13/PPV-vaccinated, n = 52) or no PPV (PCV10/PPV-naive, n = 57; PCV13/PPV-naive, n = 48). All children received a micro-dose of PPV at 23 months of age to study the capacity to respond to a pneumococcal challenge. PPV vaccination resulted in significantly increased IgG responses (1.4 to 10.5-fold change) at 10 months of age for all PPV-serotypes tested. Both PPV-vaccinated and PPV-naive children responded to the 23-month challenge and post-challenge seroprotection rates (IgG ≥ 0.35 μg/mL) were similar in the two groups (80–100% for 12 of 14 tested vaccine serotypes). These findings show that PPV is immunogenic in 9-month-old children at high risk of pneumococcal infections and does not affect the capacity to produce protective immune responses. Priming with currently available PCVs followed by a PPV booster in later infancy could offer improved protection to young children at high risk of severe pneumococcal infections caused by a broad range of serotypes.

## 1. Introduction

*Streptococcus pneumoniae* (the pneumococcus) remains a leading cause of death in children under 5 years of age and is estimated to cause over 500,000 deaths and nearly 14 million episodes of the disease annually, mainly in young children in low-income countries [1]. The epidemiology of pneumococcal infections is different in high-risk compared to low-risk settings, including that the onset and burden of pneumococcal colonization and disease happen at a younger age, often within weeks after birth, and that the spectrum of colonizing and invading pneumococcal serotypes is broader [1,2,3,4]. Preventing pneumococcal disease in children in high-risk settings requires strategies that are tailored towards providing the earliest possible protection against the broadest possible spectrum of invasive pneumococcal serotypes, and that are highly effective for at least the first 12–18 months of life when the burden of disease and death from *S. pneumoniae* is highest.

Infants in Papua New Guinea (PNG), experience one of the highest rates of pneumococcal infections worldwide. We have recently shown in a head-to-head study that the two currently available pneumococcal conjugate vaccines (PCV), the 10-valent (PCV10) and 13-valent (PCV13) vaccines, are comparably safe and immunogenic in PNG infants when given at 1, 2 and 3 months of age in line with national guidelines [5]. However, while more than 90% of infants vaccinated as part this trial produced seroprotective antibody levels against most vaccine serotypes one month after the 3rd dose of PCV10 or PCV13, antibody levels waned rapidly between 4 and 9 months of age. Giving a booster dose of PCV in later infancy may help to sustain protective antibody levels over a longer period; however, a 3+1 PCV immunization schedule may be too expensive to implement in low-income countries. An alternative is to complement priming with 3 doses of PCV with one dose of the 23-valent pneumococcal polysaccharide vaccine (PPV); this approach may not only elevate the waning antibody titers but may also induce protection against a broader spectrum of serotypes during the most critical period of life.

A 3PCV+PPV vaccination strategy was used in Australia to increase protection in high-risk Aboriginal children; however, the PPV booster, recommended at 2 years of age, was halted after a study conducted in Fiji raised concerns that PPV may deplete serotype-specific memory B-cells and limit the capacity of children to respond adequately to a pneumococcal exposure [6]. At the time of the Fiji study, we were conducting a trial in PNG that confirmed that PPV was safe and immunogenic when given to 9-month-old PNG infants (primed with 3 doses of the 7-valent PCV) [7,8]. Earlier studies in PNG, conducted before PCVs became available, had already shown that despite the limited immunogenicity of PPV in children under 2 years of age, PNG children aged 6 months to 5 years had reduced mortality and severe morbidity due to acute lower respiratory infections (ALRI) if they had been vaccinated with PPV [9,10]. Responding to the concerns raised by the Fiji study, we followed up infants vaccinated as part of the PNG PCV7/PPV trial and found that at age 3–5 years all children responded to a pneumococcal challenge with increased antibody responses [11]. While this suggests that there was no evidence of hyporesponsiveness in the PPV vaccinated PNG children, the study had two limitations. The first was the lack of a control group of children not vaccinated with PPV at 9 months of age. The second was that, as a later study reported that there was no longer evidence of hyporesponsiveness when the PPV-vaccinated Fijian children were 5–7 years old [12], follow-up of the children in the PNG study at 3-5 years of age may have been too late to show hyporesponsiveness, if any did occur.

Addressing these limitations and adding further evidence to whether a PPV booster is safe and improves levels of immune protection when given to children in high-risk settings, a second objective of the head-to-head PCV10 and PCV13 trial in PNG infants was to study the immunogenicity of PPV given at 9 months of age, persistence of antibody responses, and ability to respond to a pneumococcal challenge at 23 months of age in children who had received PPV or not. This is the first study assessing the safety and immunogenicity of a PPV booster in high-risk infants who are primed in infancy with PCV10 or PCV13.

## 2. Materials and Methods

### 2.1. Study Design

A detailed protocol of this study has been published, describing the trial aims, objectives, design, study population, and methods including consenting procedures, monitoring of adverse events, and sample size calculations [13]. The trial consisted of two parts. The aim of the first part (results reported elsewhere [5]) was to assess safety, immunogenicity and antibody persistence after PCV10 or PCV13 vaccination at 1, 2 and 3 months of age during the first 9 months of life in Papua New Guinean infants. The objectives of the second part of this study, reported here, include (1) assessing the immunogenicity and antibody persistence of PPV given at 9 months of age in children primed with PCV10 or PCV13, and (2) assessing the capacity of PPV-vaccinated compared to PPV-naive children to respond to a pneumococcal exposure by measuring IgG antibody responses produced in response to a pneumococcal challenge given in the form of a micro-dose of PPV (1/5th the normal dose) at 23 months of age.

The study was conducted according to Declaration of Helsinki International Conference on Harmonisation Good Clinical Practice (ICH-GCP) and local ethical guidelines. Ethical approval was obtained from the PNG Medical Research Advisory Committee (#11·03) and the PNG Institute of Medical Research (PNGIMR) Institutional Review Board (#1028). The study is registered with ClinicalTrials.gov (CTN NCT01619462).

### 2.2. Study Population

Between November 2011 and April 2014, 262 infants (28–35 days old) were enrolled in the 10v13vPCV trial and randomized 1:1:1:1 to receive 3 doses of PCV10 or PCV13 at 1, 2 and 3 months of age with or without a booster dose of PPV at 9 months of age [13]. The PPV group allocation was assigned at enrolment, at the same time as PCV randomization, but kept in an unopened envelope until the 9-month visit. Of the 262 enrolled children, 208 were followed-up at 9 months of age: 51 primed with PCV10 received PPV; 52 primed with PCV13 received PPV; 57 primed with PCV10 did not receive PPV; 48 primed with PCV13 did not receive PPV.

### 2.3. Study Vaccines

PCV10 (Synflorix^®^, GSK, Rixensart, Belgium, batch numbers ASPNA0099AB and ASPNA267DD) contains pneumococcal serotypes 1, 4, 5, 6B, 7F, 9V, 14 and 23F polysaccharide conjugated to non-typeable *Haemophilus influenzae* Protein D, and serotypes 18C and 19F polysaccharide conjugated to tetanus and diphtheria toxoids, respectively. PCV13 (Prevenar13^®^, Pfizer, New York City, NY, USA, batch numbers F36226 and G71540) contains pneumococcal serotypes 1, 3, 4, 5, 6A, 6B, 7F, 9V, 14, 18C, 19A, 19F and 23F conjugated to non-toxic diphtheria CRM_197_ protein. Each 0.5 mL dose of PPV (Pneumovax 23^™^, Merck & Co., Kenilworth, NJ, USA, batch numbers T0861, V1200 and K006913) contains 25 µg of purified capsular polysaccharides of serotypes 1, 2, 3, 4, 5, 6B, 7F, 8, 9N, 9V, 10A, 11A, 12F, 14, 15B, 17F, 18C, 19A, 19F, 20, 22F, 23F and 33F in 0.25% phenol preservative.

### 2.4. Study Procedures and Specimen Collection

Prior to any vaccination, the child’s medical history was assessed, and a physical examination was performed to exclude contraindications for vaccination. At age 9 months children received their second dose of measles vaccine and Vitamin A, with or without a dose of PPV [13]. All participants were given 1/5th the normal dose (0.1 mL) of PPV at age 23 months as a pneumococcal challenge. Venous blood samples (3–5 mL) were collected at 9, 10, 23 and 24 months of age.

### 2.5. Immunogenicity Assessment

Serum IgG antibodies against PCV13 serotypes and non-PCV serotype 2 were measured using the WHO standardized pneumococcal enzyme-linked immunosorbent assay (ELISA), using as described previously the human pneumococcal standard reference serum 007sp [14], and 10 μg/mL of cell wall polysaccharide (CPS) and 5 μg/mL of purified serotype 22F polysaccharide for pre-absorbance of samples to remove non-specific antibodies and increase the specificity of the assay [15,16,17]. Serotype-specific serum IgG geometric mean concentrations (GMCs), and the proportion of children with concentrations ≥0.35 μg/mL (considered the serological correlate of protection against invasive pneumococcal disease (IPD)) were calculated for each time point [18].

### 2.6. Safety Assessment

Children were observed at the clinic for 1 hour for immediate local and systemic reactions and visited at home 24–48 h postvaccination to assess local or systemic side effects. Children were followed for illness through passive surveillance throughout the study with a pre-specified analysis of the frequency of moderate and severe ALRI as previously defined [13]. A serious adverse event (SAE) was defined as any event requiring hospitalization or resulting in death.

### 2.7. Statistical Methods

Data were analysed based on an intention-to-treat analysis using SPSS 15.0 (IBM Corp., Armonk, NY, USA). Antibody concentrations were log-transformed and geometric means calculated. For all outcomes, 95% confidence intervals were calculated. A multivariate linear regression analysis adjusting for baseline IgG levels at 23 months of age was conducted to assess the relative change in IgG responses between 24 months and 23 months of age in PPV-vaccinated compared to PPV-naive children. Morbidity incidence rates and 95% confidence intervals were calculated using standard formulae.

## 3. Results

### 3.1. Study Population

Seven of the 208 children were lost to follow up between 9 and at 10 months of age: one from the PPV-vaccinated group (primed with PCV13), and six from the PPV-naive group (4 PCV10 primed, and 2 PCV13 primed). Ninety-four of the 103 PPV-vaccinated children (91%) (46 PCV10 primed and 48 PCV13 primed), and 88 of the 105 the children in the PPV-naive group (84%) (48 PCV10 primed and 40 PCV13 primed) were given a 1/5th challenge dose of PPV at 23 months of age. Antibody responses were assessed one month later for 89 (95%) of the PPV-vaccinated children and 87 (99%) of the PPV-naive children. Reasons for loss to follow-up at different time points were reported previously [13].

### 3.2. Reactogenicity, Safety and Morbidity

Reactogenicity was minimal, with one child (PCV13-primed) presenting with tenderness at the PPV vaccination site, one child presenting with diarrhea and another child with irritability (both PCV10-primed) within 48 h after PPV and measles vaccination. None of the children in the PPV-naive group presented with any systemic side effects after measles vaccination. 

A total of 331 illness episodes were documented between 9 and 24 months of age. No vaccine-related SAEs occurred. Five children died of malnutrition or gastroenteritis, all between 15 and 19 months of age: three in the PPV-naive and two in the PPV-vaccinated group. Incidence rates of any morbidity, any ALRI, moderate/severe ALRIs, and all-cause hospitalization are presented in Table 1. The incidence of all-cause hospitalization was lower in PPV-vaccinated than in PPV-naive children (incidence rate ratio (IRR) is 0.26, 0.07–0.61). There were no significant differences in the incidence of all-cause morbidity (IRR is 0.99, 0.80–1.24), any ALRI (IRR is 1.06, 0.73–1.56), or moderate/severe ALRI (IRR is 1.29, 0.70–2.47) between PPV-vaccinated and PPV-naive groups between 9 and 24 months of age, nor when analyzing data for 9 to 12 months, and 12 to 24 months of age, separately (data not shown). These data support that PPV is safe and not associated with increased risk for disease.

### 3.3. PPV Immunogenicity and Seroprotection Rates at 10 Months of Age in PCV10- and PCV13-Primed Children

Vaccination with PPV at 9 months of age resulted in significantly increased IgG responses (1.4 to 10.5-fold increase) at 10 months of age for all PPV-serotypes tested (Table 2) (Appendix A). For serotype 6A, which is not contained in PPV, IgG responses increased in the PPV-vaccinated group (1.6-fold increase in both the PCV10- and PCV13-primed group), possibly due to cross-reactivity with serotype 6B. GMCs for all measured serotype-specific IgG antibodies were similar one month after PPV vaccination in children primed with PCV10 or PCV13, except for higher serotype 19A responses in the PCV13-primed than in the PCV10-primed group (Table 2). In the PPV-naive group IgG responses remained the same or declined between 9 and 10 months of age at the same rate for PCV10- and PCV13-primed children. These findings demonstrate that PPV is immunogenic in 9-month-old infants who have been primed with PCV10 or PCV13.

Based on IgG ≥ 0.35 µg/mL, seroprotection rates one month after PPV vaccination varied between 92% and 100% for all PPV serotypes, regardless of whether children were primed with PCV10 or PCV13. Compared to PPV-naive children, seroprotection rates were higher in PPV-vaccinated children for all PPV serotypes, except for serotypes for which seroprotection rates were also high in the PPV-naive group such as 7F, 14, 19A and 19F. In the PPV-naive group seroprotection rates at 10 months of age for shared PCV10/PCV13 serotypes varied between 44% (PCV13) and 59% (PCV10) for serotype 4, and 45% (PCV10) and 51% (PCV13) for serotype 23F, to 96% (PCV10) and 98% (PCV13) for serotype 14 (Figure 1 and Appendix A). In the PPV-naive group, seroprotection rates for serotypes included in PCV13 but not PCV10 were higher for serotype 6A in the PCV13 (80%, 68–92) than PCV10 group (33%, 20–45), but did not differ for serotypes 3 and 19A (Figure 1 and Appendix A).

Using a more conservative cut-off of IgG levels ≥ 1.0 μg/mL, at 10 months of age more than 85% of PPV-vaccinated children had levels above this cut-off for 8 of the 10 PCV10/ PCV13 serotypes, and 65% for the other two serotypes 4 and 23F (Figure 1 and Appendix A). Within the PPV-vaccinated group, at 10 months of age a higher proportion of PCV13-primed than PCV10-primed children had IgG levels ≥ 1.0 μg/mL for the PCV13-only serotypes 6A (PCV13: 31%, 18–44) (PCV10: 14%; 4–24) and 19A (PCV13: 92%; 84–100) (PCV10: 57%; 43–71), but proportions were comparable for serotype 3 (in both groups 55%; 41–69).

These findings show that the persistence of seroprotection induced by priming with 3 doses of PCV10 or PCV13 in early infancy varies between serotypes and that the administration of a booster dose of PPV in later infancy improves levels of seroprotection.

### 3.4. Persistence of Serotype-Specific IgG Responses at 23 Months of Age

IgG responses declined between 10 and 23 months of age in children vaccinated with PPV (Appendix A). While in the PCV10-primed group GMCs for IgG against serotypes shared between PCV10 and PCV13 remained higher in the PPV-vaccinated than in the PPV-naive group for 8 of the 10 serotypes (serotypes 1, 4, 5, 7F, 9V, 18C, 19F and 23F), in the PCV13-primed group antibody levels at 23 months of age were similar for children who had or had not received PPV (Table 3). IgG GMCs for all shared PCV10 and PCV13 serotypes were, however, comparable at 23 months of age between PCV10-primed and PCV13-primed children, vaccinated with PPV at 9 months of age.

At 23 months of age seroprotection rates based on IgG levels ≥ 0.35 µg/mL for 5 of the 10 shared PCV10 and PCV13 serotypes (1, 4, 9V, 18C and 23F), and based on IgG levels ≥ 1.0 µg/mL for serotype 19F were higher in PPV-vaccinated than in PPV-naive children when primed in early infancy with PCV10 (Figure 2 and Appendix A). For PCV13-primed children seroprotection rates at 23 months of age were similar in those who did or did not receive PPV at 9 months of age. Seroprotection rates at 23 months of age did, however, not differ between children primed with PCV10 or PCV13.

Overall, these data indicate that in the 14 months since vaccination, PPV-induced IgG antibodies have waned to pre-vaccination levels comparable to children who did not receive PPV.

### 3.5. Response of PPV-Vaccinated and PPV-Naive Children to a Pneumococcal Challenge at 23 Months of Age

All children responded to a pneumococcal challenge given in the form of a 1/5th dose of PPV at 23 months of age as shown by higher serotype-specific IgG antibody responses one month later (Table 3 and Appendix A). However, as demonstrated in a multivariate regression analysis that adjusted for the level of IgG antibodies before PPV challenge at 23 months of age (Table 3), this response was smaller in PPV-vaccinated children than in PPV-naive children for serotypes 1, 2, 6B, 7F, 9V, and 18C in both PCV10- and PCV13-primed children, for serotypes, 4, 5, 6A 23F in PCV13-primed children, and for serotype 19F in PCV10-primed children. Nevertheless, most PPV-vaccinated and PPV-naive children produced IgG antibody above the seroprotective titer after the challenge: in both the PPV-vaccinated and PPV-naive group more than 80% of children had IgG levels ≥ 0.35 µg/mL for all the tested PPV serotypes except for serotype 3 (Appendix A).

This confirms that 14 months after PPV vaccination, children produce levels of IgG antibodies in response to a pneumococcal challenge that is considered protective; however, memory B-cell responses may at this time not yet have been fully restored.

## 4. Discussion

This study confirms earlier findings that PPV is immunogenic and induces high levels of seroprotection when given under the age of 12 months to PCV-primed children in high-risk settings [8,19]. We also show that 14 months after vaccination, when PPV-induced IgG antibodies have waned, children can respond to a pneumococcal challenge by producing seroprotective levels of serotype specific IgG.

More specifically, we found that PPV given to 9-month-old PNG infants primed with PCV10 or PCV13 resulted in a 2- to 10-fold increase in circulating serotype-specific IgG antibodies; a nearly 100% seroprotection based on IgG responses ≥ 0.35 μg/mL for most PCV serotypes and non-PCV serotype 2; and at least 80% of children achieved IgG responses ≥ 1 μg/mL. Fourteen months after PPV vaccination more than 80% children still had seroprotective antibody levels for most serotypes, although IgG GMCs had decreased to near pre-PPV vaccination levels and were comparable to those in children who had not received PPV. Serotypes 3 and 4 had the most rapid waning and lowest seroprotection rates. At this time, both PPV-vaccinated and PPV-naive children responded to a pneumococcal challenge: PPV-naïve children with a 1.5- to 4-fold increase in vaccine-serotype IgG and PPV-vaccinated children with a maximum 2-fold increase.

These are relevant findings considering that in high-risk settings most pneumococcal disease and deaths occur in the first two years of life, and the coverage afforded by PCV10 and PCV13 is limited due to the broad range of pneumococcal serotypes colonizing and causing disease (for example, less than 50% of IPD serotypes in PNG are contained in PCV13) [3,4,5]. PCV10 and PCV13, like the earlier 7-valent vaccine, are also associated with an increase in disease due to non-vaccine serotypes [20], which in particular in high-risk settings could reduce the impact afforded by these vaccines. Until new vaccines become available, complementing priming with PCV with one dose of PPV to increase antibody titers and serotype coverage during the two most critical years of life may, therefore, be a strategy to protect children in high-risk settings in the first two critical years of life. 

While PPV does not induce immunological memory and its immunogenicity in young infants has limitations [21], PCVs are highly immunogenic in infants, induce immunological memory and antibodies with opsonophagocytic activity, and are effective in preventing IPD caused by vaccine serotypes in young children in low and high-risk countries [22,23,24,25]. Nevertheless, as shown here and by other studies, PPV is immunogenic and can induce high avidity, opsonophagocytic antibodies when given to PCV-primed infants under the age of 2 in high-risk settings and as young as 9 months of age in PNG [8,19,26,27]. Importantly, early studies conducted in PNG before PCVs became available have shown that PPV had a 45% protective effect against moderate/severe ALRI at the time of an epidemic [9]; a 50–59% efficacy against ALRI mortality [10]; and reduced overall mortality [28], hence suggesting that these PPV-induced immune responses are protective. That PPV can induce a certain level of protective immune responses in young infants in PNG could possibly be explained by the high level of pneumococcal exposure experienced by these young infants.

Despite the benefits that PPV vaccination can offer to young children in high-risk settings, findings by a Fiji study that children vaccinated at 12 months of age with PPV failed to boost their antibody responses to a pneumococcal challenge 5 months later (by giving children a 1/5th dose of PPV as we did in this study) escalated concerns that PPV may lead to transient immune hyporesponsiveness [6]. Long-term protection against pneumococci requires the induction and persistence of memory B-cells that can replace plasma cells and produce protective antibodies in response to a challenge [29]: concerns are that PPV depletes this memory B-cell compartment by activating memory B-cells to differentiate into antibody-secreting plasma cells [30,31,32]. Since Fijian study participants were challenged only 5 months after they were vaccinated with PPV, it is possible that the high circulating antibody levels that were still present in the PPV-vaccinated group may have limited the challenge response. Or that in this population with relatively lower carriage rates, there was insufficient time for new pneumococcal exposures to regenerate the memory B-cell pool. We show here that 14 months after PPV vaccination when vaccine-induced antibodies have waned, children responded to a pneumococcal challenge with producing seroprotective antibody levels against most PCV serotypes and serotype 2. However, as the magnitude of the response was smaller in children who had received PPV, this indicates that B-cell memory in this group is not yet fully restored. While there was no evidence that children were at increased risk of disease at any time in the 14 months after PPV immunization, it will be important to confirm the functional activity of the antibodies circulating at that time and whether there are any effects on vaccine serotype pneumococcal carriage and bacterial load.

A limitation of conducting a field trial as intensive as this one under challenging logistical conditions and with limited funding is that the size of the cohort that can be studied is restricted. As just under 80% of children remained in the study at 9 months of age (most losses to follow-ups occurred in the first 3 months of the study [13]), and another 32 children (including 5 deaths) were lost to follow up between 10 and 24 months of age, the number of children that completed the follow-up after the challenge dose was relatively low but still adequate to study the effects of PPV on safety and immune responsiveness. Another limitation is that we did not study the kinetics of antibody waning in the interval between 9 and 23 months of age, although the high seroprotection rates at 23 months of age suggest that children were protected in the interim. Furthermore, it will be important to demonstrate the presence of serotype-specific memory B-cells. To this end, we are planning to conduct B-cell studies, and opsonophagocytosis assays to confirm the functional capacity of the antibodies produced following PPV vaccination.

In summary, this study indicates that until PCVs with broader serotype coverage or serotype-independent vaccines become available, priming with currently available PCVs in early infancy followed by a PPV booster in late infancy is likely to be the best available strategy to protect young children in high-risk settings against severe pneumococcal infections caused by a broad range of pneumococcal serotypes.

## Figures and Tables

**Figure 1 vaccines-07-00017-f001:**
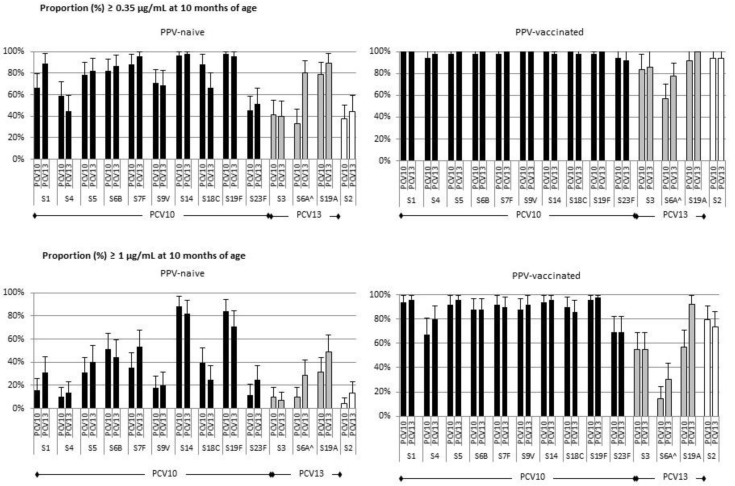
Proportion of children with serotype-specific IgG antibodies ≥ 0.35 μg/mL or ≥ 1.0 μg/mL at 10 months of age. Infants primed in early infancy with PCV10 or PCV13 received one dose of PPV or no PPV at 9 months of age. Serotype-specific IgG antibodies were measured one month later. ^ Serotype 6A is not included in PPV.

**Figure 2 vaccines-07-00017-f002:**
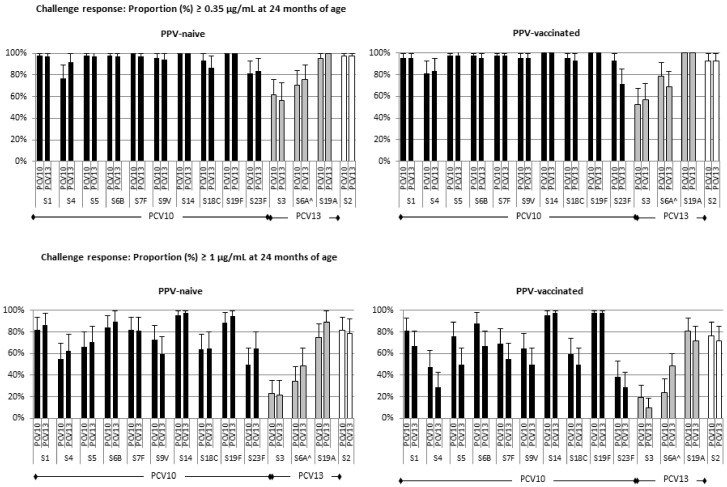
Proportion of children with serotype-specific IgG antibodies ≥ 0.35 μg/mL or ≥ 1.0 μg/mL at 23 months of age. The presence of seroprotective serotype-specific IgG responses was assessed 14 months after children did (PPV-vaccinated) or did not (PPV-naïve) receive one dose of PPV at 9 months of age. All children had received 3 doses of PCV10 or PCV13 at 1, 2 and 3 months of age. ^ Serotype 6A is not included in PPV.

**Table 1 vaccines-07-00017-t001:** Incidence rates of any morbidity, hospitalization, and any or moderate/severe acute lower respiratory tract infections (ALRI) between 9 and 24 months of age in 23-valent pneumococcal polysaccharide vaccine (PPV)-vaccinated and PPV-naive children.

Adverse Event	PPV-Naive				PPV-Vaccinated			
	PCV10-Primed		PCV13-Primed		PCV10-Primed		PCV13-Primed	
	Episodes	Incidence (ppy) (95% CI)	Episodes	Incidence (ppy) (95% CI)	Episodes	Incidence (ppy) (95% CI)	Episodes	Incidence (ppy) (95% CI)
Any morbidity	79	1.27 (1.02–1.59)	84	1.54 (1.24–1.90)	85	1.42 (1.15–1.76)	83	1.36 (1.10–1.68)
Any ALRI	25	0.40 (0.27–0.60)	26	0.48 (0.32–0.70)	30	0.50 (0.35–0.72)	26	0.43 (0.29–0.62)
Moderate/severe ALRI	12	0.19 (0.11–0.34)	6	0.11 (0.05–0.24)	13	0.22 (0.13–0.37)	11	0.18 (0.10–0.33)
Hospitalizations	7	0.11 (0.05–0.24)	4	0.07 (0.03–0.19)	0	0 (0–0)	3	0.05 (0.02–0.15)

Incidence is per person-year (ppy).

**Table 2 vaccines-07-00017-t002:** Immunogenicity of 23-valent pneumococcal polysaccharide vaccine (PPV) booster at 9 months of age: serotype-specific IgG responses in PPV-vaccinated and PPV-naive children at 9 and 10 months of age according to priming with PCV10 or PCV13 in early infancy.

Serotypes	Priming	PPV-Naive			PPV-Vaccinated		
		GMC IgG (µg/mL) (95% CI)		GM Fold Change (95% CI)	GMC IgG (µg/mL) (95% CI)		GM Fold Change (95% CI)
		9 Months	10 Months		9 Months	10 Months	
PCV10/PCV13 serotypes							
1	PCV10	0.57 (0.46–0.70)	0.51 (0.38–0.68)	0.86 (0.69–1.08)	0.70 (0.56–0.88)	7.52 (5.62–10.06)	10.45 (7.92–13.79)
	PCV13	0.98 (0.76–1.27)	0.79 (0.60–1.04)	0.80 (0.70–0.90)	0.83 (0.70–0.98)	5.65 (4.25–7.51)	6.82 (5.19–8.97)
4	PCV10	0.42 (0.34–0.52)	0.39 (0.32–0.49)	0.90 (0.78–1.03)	0.48 (0.40–0.59)	1.68 (1.27–2.23)	3.37 (2.63–4.30)
	PCV13	0.43 (0.34–0.55)	0.37 (0.28–0.48)	0.84 (0.70–1.01)	0.39 (0.32–0.46)	2.44 (1.89–3.17)	6.40 (4.97–8.25)
5	PCV10	0.67 (0.56–0.82)	0.64 (0.51–0.80)	0.91 (0.79–1.04)	0.64 (0.52–0.78)	3.09 (2.45–3.91)	4.91 (3.81–6.33)
	PCV13	0.87 (0.69–1.10)	0.72 (0.55–0.94)	0.81 (0.69–0.94)	0.77 (0.64–0.93)	2.65 (2.15–3.26)	3.38 (2.63–4.34)
6B	PCV10	1.00 (0.80–1.25)	0.94 (0.74–1.19)	0.97 (0.87–1.09)	1.18 (0.93–1.51)	3.85 (2.86–5.18)	3.19 (2.56–3.96)
	PCV13	0.96 (0.73–1.27)	0.88 (0.66–1.18)	0.91 (0.74–1.11)	0.77 (0.63–0.95)	4.08 (3.05–5.45)	5.40 (3.99–7.31)
7F	PCV10	0.77 (0.65–0.92)	0.73 (0.59–0.90)	0.93 (0.82–1.05)	0.81 (0.66–1.00)	3.21 (2.54–4.05)	3.82 (3.10–4.70)
	PCV13	1.21 (0.97–1.51)	1.01 (0.81–1.26)	0.83 (0.72–0.95)	0.92 (0.76–1.10)	3.15 (2.52–3.96)	3.45 (2.77–4.31)
9V	PCV10	0.54 (0.46–0.65)	0.50 (0.39–0.64)	0.87 (0.76–1.01)	0.62 (0.52–0.74)	4.45 (3.36–5.89)	7.04 (5.26–9.41)
	PCV13	0.66 (0.54–0.80)	0.48 (0.39–0.59)	0.78 (0.65–0.93)	0.48 (0.39–0.59)	3.32 (2.60–4.25)	7.13 (5.45–9.34)
14	PCV10	3.05 (2.45–3.78)	2.64 (2.06–3.37)	0.85 (0.74–0.98)	2.93 (2.16–3.98)	9.11 (6.39–12.98)	3.05 (2.11–4.40)
	PCV13	4.09 (2.99–6.60)	3.31 (2.44–4.49)	0.81 (0.72–0.90)	3.49 (2.63–4.65)	6.39 (4.63–8.82)	1.86 (1.53–2.28)
18C	PCV10	1.02 (0.82–1.27)	0.89 (0.71–1.11)	0.84 (0.75–0.93)	0.89 (0.71–1.11)	2.69 (2.06–3.51)	2.90 (2.24–3.75)
	PCV13	0.66 (0.54–0.81)	0.51 (0.41–0.64)	0.77 (0.69–0.86)	0.63 (0.52–0.78)	2.82 (2.16–3.67)	4.71 (3.59–6.17)
19F	PCV10	2.39 (1.91–3.00)	2.52 (1.96–3.23)	1.02 (0.90–1.16)	2.30 (1.86–2.86)	7.35 (5.45–9.93)	3.10 (2.47–3.89)
	PCV13	1.96 (1.51–2.56)	1.73 (1.31–2.30)	0.90 (0.76–1.06)	1.58 (1.25–1.99)	9.50 (7.41–12.18)	6.25 (4.73–8.27)
23F	PCV10	0.38 (0.30–0.49)	0.36 (0.28–0.46)	0.91 (0.74–1.13)	0.47 (0.38–0.60)	1.63 (1.20–2.22)	3.40 (2.57–4.49)
	PCV13	0.45 (0.34–0.60)	0.38 (0.27–0.52)	0.84 (0.69–1.02)	0.42 (0.32–0.56)	2.08 (1.52–2.83)	5.24 (3.68–7.47)
PCV13 serotypes							
3	PCV10	0.33 (0.24–0.44)	0.29 (0.22–0.37)	0.87 (0.76–1.00)	0.36 (0.27–0.48)	0.94 (0.70–1.27)	2.38 (1.79–3.17)
	PCV13	0.30 (0.21–0.41)	0.28 (0.21–0.38)	0.93 (0.80–1.07)	0.30 (0.24–0.37)	1.05 (0.83–1.35)	3.61 (2.80–4.67)
6A^	PCV10	0.24 (0.19–0.30)	0.24 (0.18–0.31)	0.95 (0.85–1.07)	0.27 (0.21–0.35)	0.45 (0.34–0.58)	1.56 (1.32–1.84)
	PCV13	0.64 (0.47–0.88)	0.59 (0.44–0.80)	0.92 (0.76–1.12)	0.42 (0.33–0.52)	0.67 (0.53–0.85)	1.61 (1.31–1.98)
19A	PCV10	0.81 (0.65–1.02)	0.79 (0.61–1.02)	0.96 (0.85–1.09)	0.91 (0.73–1.14)	1.31 (0.93–1.83)	1.41 (1.17–1.70)
	PCV13	1.32 (0.94–1.84)	1.02 (0.71–1.46)	0.80 (0.66–0.97)	1.14 (0.88–1.46)	3.80 (2.80–5.15)	3.38 (2.46–4.63)
Non-PCV serotype							
2	PCV10	0.30 (0.25–0.37)	0.29 (0.24–0.35)	0.94 (0.74–1.21)	0.34 (0.27–0.44)	2.05 (1.51–2.78)	5.96 (4.33–8.22)
	PCV13	0.38 (0.24–0.43)	0.32 (0.25–0.43)	0.84 (0.67–1.06)	0.30 (0.26–0.35)	1.80 (1.35–2.39)	6.02 (4.45–8.15)

^ Serotype 6A is not included in PPV.

**Table 3 vaccines-07-00017-t003:** Persistence, fold-change and relative fold-change of vaccine serotype-specific IgG antibody responses (GMCs) at 23 months and 24 months of age in 23-valent pneumococcal polysaccharide vaccine (PPV)-vaccinated and PPV-naive children, and relative fold-change in PPV-vaccinated versus PPV-naive children.

Serotypes	Priming	23 Months IgG GMC (µg/mL) (95% CI)		24 Months IgG GMC (µg/mL) (95% CI)		Fold-Change 24 Months/23 Months (95% CI)		Relative Fold-Change (95% CI)
		PPV-Naive	PPV-Vaccinated	PPV-Naive	PPV-Vaccinated	PPV-Naive	PPV-Vaccinated	PPV-Vaccinated/PPV-Naive
PCV10/PCV13 serotypes								
1	PCV10	0.58 (0.43–0.79)	1.06 (0.81–1.39)	2.45 (1.88–3.20)	1.89 (1.43–2.51)	4.13 (2.87–5.95)	1.85 (1.49–2.28)	0.62 (0.43–0.89)
	PCV13	0.70 (0.49–1.02)	0.85 (0.65–1.11)	2.51 (1.85–3.42)	1.39 (1.04–1.86)	3.49 (2.46–4.95)	1.76 (1.40–2.21)	0.53 (0.37–0.76)
4	PCV10	0.29 (0.23–0.38)	0.47 (0.39–0.57)	0.86 (0.63–1.16)	0.85 (0.66–1.09)	2.81 (2.07–3.81)	1.84 (1.46–2.32)	0.79 (0.55–1.15)
	PCV13	0.41 (0.29–0.58)	0.37 (0.29–0.47)	1.34 (0.97–1.85)	0.67 (0.54–0.83)	3.23 (2.27–4.59)	1.88 (1.50–2.36)	0.54 (0.39–0.76)
5	PCV10	0.68 (0.54–0.85)	1.00 (0.81–1.23)	1.62 (1.29–2.05)	1.45 (1.15–1.83)	2.34 (1.76–3.11)	1.45 (1.15–1.82)	0.77 (0.56–1.07)
	PCV13	0.86 (0.65–1.14)	0.86 (0.70–1.06)	1.73 (1.30–2.32)	1.08 (0.87–1.33)	2.02 (1.59–2.57)	1.28 (1.09–1.50)	0.63 (0.49–0.82)
6B	PCV10	0.98 (0.76–1.25)	1.39 (1.11–1.74)	2.93 (2.24–3.83)	2.34 (1.80–3.03)	2.88 (2.16–3.85)	1.72 (1.36–2.17)	0.70 (0.50–0.99)
	PCV13	1.26 (0.93–1.71)	1.00 (0.76–1.32)	3.93 (3.01–5.13)	1.31 (1.02–1.69)	3.15 (2.35–4.23)	1.39 (1.13–1.70)	0.39 (0.29–0.53)
7F	PCV10	0.72 (0.57–0.90)	1.12 (0.93–1.34)	1.85 (1.48–2.32)	1.55 (1.25–1.92)	2.52 (1.95–3.25)	1.37 (1.18–1.60)	0.67 (0.51–0.89)
	PCV13	0.98 (0.76–1.27)	0.83 (0.65–1.04)	2.19 (1.69–2.85)	1.24 (1.01–1.51)	2.18 (1.77–2.68)	1.53 (1.30–1.79)	0.66 (0.52–0.83)
9V	PCV10	0.45 (0.35–0.58)	0.84 (0.68–1.04)	1.64 (1.27–2.11)	1.30 (1.06–1.61)	3.53 (2.51–4.98)	1.57 (1.31–1.87)	0.66 (0.47–0.92)
	PCV13	0.62 (0.45–0.85)	0.68 (0.55–0.85)	1.51 (1.13–2.01)	1.06 (0.85–1.31)	2.36 (1.79–3.11)	1.58 (1.26–1.98)	0.69 (0.51–0.94)
14	PCV10	2.37 (1.88–2.99)	3.07 (2.55–3.70)	4.92 (3.63–6.67)	5.69 (4.21–7.68)	2.00 (1.35–2.97)	1.83 (1.40–2.38)	1.08 (0.70–1.67)
	PCV13	2.71 (2.05–3.58)	2.08 (1.66–2.62)	6.09 (4.49–8.24)	4.33 (3.36–5.58)	2.13 (1.53–2.97)	2.12 (1.67–2.69)	0.85 (0.59–1.22)
18C	PCV10	0.44 (0.35–0.56)	0.60 (0.50–0.73)	1.51 (1.15–1.98)	1.07 (0.85–1.35)	3.31 (2.33–4.70)	1.77 (1.43–2.19)	0.67 (0.47–0.95)
	PCV13	0.55 (0.42–0.71)	0.66 (0.52–0.84)	1.40 (1.03–1.91)	1.05 (0.83–1.33)	2.49 (1.84–3.35)	1.60 (1.31–1.95)	0.70 (0.50–0.96)
19F	PCV10	1.51 (1.19–1.91)	2.32 (1.87–2.87)	5.14 (3.64–7.26)	4.21 (3.28–5.39)	3.31 (2.42–4.52)	1.83 (1.46–2.30)	0.64 (0.43–0.94)
	PCV13	2.40 (1.78–3.21)	2.27 (1.72–2.99)	6.10 (4.25–8.76)	4.69 (3.50–6.29)	2.67 (2.00–3.55)	2.12 (1.70–2.64)	0.79 (0.56–1.12)
23F	PCV10	0.36 (0.28–0.47)	0.52 (0.42–0.64)	0.89 (0.69–1.15)	0.84 (0.68–1.05)	2.39 (1.72–3.32)	1.62 (1.29–2.03)	0.86 (0.62–1.21)
	PCV13	0.52 (0.40–0.69)	0.45 (0.34–0.60)	1.31 (0.94–1.84)	0.61 (0.46–0.82)	2.54 (1.90–3.41)	1.38 (1.11–1.71)	0.52 (0.37–0.73)
PCV13 serotypes								
3	PCV10	0.24 (0.20–0.30)	0.26 (0.21–0.32)	0.46 (0.36–0.59)	0.41 (0.32–0.53)	1.98 (1.64–2.39)	1.60 (1.34–1.91)	0.82 (0.64–1.06)
	PCV13	0.24 (0.18–0.31)	0.26 (0.21–0.32)	0.41 (0.31–0.53)	0.40 (0.33–0.49)	1.71 (1.42–2.06)	1.55 (1.32–1.82)	0.94 (0.75–1.17)
6A^	PCV10	0.38 (0.28–0.50)	0.41 (0.33–0.51)	0.55 (0.40–0.77)	0.61 (0.47–0.81)	1.49 (1.27–1.76)	1.48 (1.19–1.84)	1.00 (0.76–1.31)
	PCV13	0.61 (0.44–0.85)	0.42 (0.33–0.55)	0.90 (0.65–1.25)	0.51 (0.39–0.66)	1.47 (1.19–1.83)	1.23 (1.04–1.45)	0.77 (0.59–1.00)
19A	PCV10	1.11 (0.88–1.40)	1.36 (1.09–1.69)	2.10 (1.50–2.93)	2.00 (1.51–2.66)	1.84 (1.39–2.44)	1.44 (1.16–1.78)	0.81 (0.57–1.16)
	PCV13	2.20 (1.60–3.01)	1.59 (1.16–2.18)	4.27 (3.05–5.99)	2.48 (1.80–3.44)	1.94 (1.41–2.66)	1.60 (1.30–1.96)	0.75 (0.53–1.06)
Non-PCV serotype								
2	PCV10	0.74 (0.57–0.95)	1.00 (0.81–1.23)	1.75 (1.43–2.14)	1.54 (1.22–1.95)	2.40 (1.89–3.05)	1.56 (1.27–1.91)	0.76 (0.58–1.00)
	PCV13	0.62 (0.48–0.79)	0.90 (0.69–1.16)	1.84 (1.43–2.36)	1.31 (1.01–1.70)	3.04 (2.32–3.97)	1.44 (1.21–1.71)	0.55 (0.41–0.73)

Serotype specific IgG antibodies were measured before and 1 month after children were challenged with a 1/5th dose of PPV at 23 months of age. The table shows geometric mean concentrations (GMCs and 95% CI) for IgG antibodies at 23 months and 24 months, fold-changes in IgG antibodies between 24 months and 23 months (geometric mean (GM) fold changes and 95% CI), and relative fold changes in 24 versus 23 month IgG antibodies between PPV-vaccinated (reference) compared to PPV-naive children as calculated in a multivariate regression analysis adjusting for antibody concentrations at 23 months (baseline before challenge). ^ Serotype 6A is not included in PPV.

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
