# Peer review of "Immunogenicity and Immune Memory after a Pneumococcal Polysaccharide Vaccine Booster in a High-Risk Population Primed with 10-Valent or 13-Valent Pneumococcal Conjugate Vaccine: A Randomized Controlled Trial in Papua New Guinean Children"

_vaccines, 2019, doi:10.3390/vaccines7010017_

Round 1

Reviewer 1 Report

General Comments:

Overall, the study is very significant as it measure important immune corelates for protection against a global disease, cause by S. pneumoniae. The authors wrote a very good manuscript and provided all necessary background information about the recent study on PCV vaccine. They also carefully included the drawbacks and limitations of the study conducted in Fiji and by them while using the PPV vaccine in PNG. 

But, there are few concerns with regards to the authors claim and report. First, I would not call it an immune memory since they are specifically measuring just the antibody response and not cell mediated immune response. Secondly, it is not clear to the reader about the challenge strategy at 10-month of age. It would be more clear if the authors mention the use of a pneumococcal strain or PPV. Lastly, does the authors think that only antibody response with a standard seroprotection amount would be consider as safe and protective? Or, will they consider a vaccine booster more effective due to a combination of antibody response and T cell mediated immune response. No doubt that the increased antigen immunized led to increased antibody response and therefore better chance for clearing the pathogen. 

Specific Comments:

Line 2, 22: Does the authors agree that the study is about  antibody response rather than immune memory.

Line 27,28: Did the authors meant challenge by the vaccine? 

Line 58, 59: How will PPV booster lower the cost? 

Line 64-66: The present study did not have data on B-cell depletion. Did the authors check it? Or, its a future plan (Line 330)?

Line 72: ALRI ? ALRI is mention first time here. Please provide the detail instead of abbreviation.

Line 130: How do the authors account for any false positive IgGs? Is there any other alternative to confirm the seroprotective features?

Author Response

Overall, the study is very significant as it measure important immune corelates for protection against a global disease, cause by S. pneumoniae. The authors wrote a very good manuscript and provided all necessary background information about the recent study on PCV vaccine. They also carefully included the drawbacks and limitations of the study conducted in Fiji and by them while using the PPV vaccine in PNG.

Thank you very much for your admiration.

But, there are few concerns with regards to the authors claim and report. First, I would not call it an immune memory since they are specifically measuring just the antibody response and not cell mediated immune response. Secondly, it is not clear to the reader about the challenge strategy at 10-month of age. It would be more clear if the authors mention the use of a pneumococcal strain or PPV. Lastly, does the authors think that only antibody response with a standard seroprotection amount would be consider as safe and protective? Or, will they consider a vaccine booster more effective due to a combination of antibody response and T cell mediated immune response. No doubt that the increased antigen immunized led to increased antibody response and therefore better chance for clearing the pathogen. 

Line 2, 22: Does the authors agree that the study is about  antibody response rather than immune memory.

Line 27,28: Did the authors meant challenge by the vaccine?

Responses to All Above:

As we believe that the five points raised by the reviewer may suggest that the reviewer may be confused about the objectives and methodology, we felt it might be better to respond to all points at the same time.

The objectives of this study were to study the immunogenicity of PPV when given to infants at 9 months old (who had received 3 prior doses of pneumococcal conjugate vaccines), and to study whether giving PPV at 9 months old may have a negative impact on the capacity to respond to pneumococci. The latter has been suggested by an earlier study conducted in Fiji: it has been suggested that PPV (a vaccine that does not induce immune memory) stimulates memory B-cells that have been induced by earlier PCV vaccination, and therefore (since PPV does not generate new memory B-cells) would deplete the memory B-cell pool and hence lead to hyporesponsiveness (a reduced capacity to respond to a pneumococcal exposure). To test whether PPV vaccination results in hyporesponsiveness, a micro-dose of PPV (1/5th the normal dose) is given (to mimic a natural exposure to pneumococci) as a pneumococcal challenge to measure the capacity to produce protective antibody responses.

Hence, PPV given at 9 months is to immunize the children and induce antibody responses to the 23 serotypes included in PPV. The micro-dose given at 23 months is not to immunize but is given as a challenge (mimic a natural exposure) to measure whether the capacity to produce protective antibody responses is retained.

The immunogenicity of PPV (given at 9 months) is assessed by measuring antibody responses 1 month later.

The capacity to respond to the challenge (given at 23 months) is assessed by measuring the capacity to produce protective antibody responses as assessed one month later. To assess whether PPV given at 9 months results in depletion of memory B-cells, it would be possible to measure serotype-specific memory B-cells; however, this has not been done but may be done so in the future.

The study does not aim to measure cellular immune responses, as PPV immunization does not induce cellular immunity.

To better describe the objectives and methodologies of the study, we have rewritten the abstract and clarified in a few sentences throughout the text that the micro-dose of PPV given at 23 months is to mimic a pneumococcal exposure.

Rewritten abstract:

"We investigated the immunogenicity, seroprotection rates and persistence of immune memory in young children at high-risk of pneumococcal disease in Papua New Guinea (PNG). Children were primed with 10-valent (PCV10) or 13-valent pneumococcal conjugate vaccines (PCV13) at 1, 2 and 3 months of age and randomized at 9 months to receive PPV (PCV10/PPV-vaccinated, n=51; PCV13/PPV-vaccinated, n=52) or no PPV (PCV10/PPV-naive, n=57; PCV13/PPV-naive, n=48). All children received a micro-dose of PPV at 23 months of age to study the capacity to respond to a pneumococcal challenge. PPV vaccination resulted in significantly increased IgG responses (1.4 to 10.5-fold change) at 10 months of age for all PPV-serotypes tested. Both PPV-vaccinated and PPV-naive children responded to the 23-month challenge and post-challenge seroprotection rates (IgG ≥ 0.35μg/mL) were similar in the two groups (80-100% for 12 of 14 tested vaccine serotypes). These findings show that PPV is immunogenic in 9-month-old children at high risk of pneumococcal infections and does not affect the capacity to produce protective immune responses. Priming with currently available PCVs followed by a PPV booster in later infancy could offer improved protection to young children at high risk of severe pneumococcal infections caused by a broad range of serotypes."

Line 58, 59: How will PPV booster lower the cost?

Response: PPV is a cheaper vaccine than PCV (that is, when procured commercially for booster vaccination).

Line 64-66: The present study did not have data on B-cell depletion. Did the authors check it? Or, its a future plan (Line 330)?

Response: Memory B-cell assays were not performed but as mentioned in the discussion are planned to be conducted in the future: “Furthermore, it will be important to demonstrate the presence of serotype-specific memory B-cells and to this end we are planning to conduct B-cell studies, and opsonophagocytosis assays to confirm the functional capacity of the antibodies produced following PPV vaccination.”

Line 72: ALRI ? ALRI is mention first time here. Please provide the detail instead of abbreviation.

Response: We have added this to the sentence, “…PNG children aged 6 months to 5 years had reduced mortality and severe morbidity due to acute lower respiratory infections (ALRI) if…”

Line 130: How do the authors account for any false positive IgGs? Is there any other alternative to confirm the seroprotective features?

Response: As mentioned in the methods we used the WHO standardized ELISA for measuring pneumococcal serotype specific IgG antibodies that includes a pre-incubation step to remove non-specific antibodies.

"Serum IgG antibodies against PCV13 serotypes and non-PCV serotype 2 were measured using the WHO standardized pneumococcal enzyme-linked immunosorbent assay (ELISA), using as described previously the human pneumococcal standard reference serum 007sp [14], and 10 μg/mL of cell wall polysaccharide (CPS) and 5 μg/mL of purified serotype 22F polysaccharide for pre-absorbance of samples to remove non-specific antibodies and increase the specificity of the assay [15-17]. Serotype-specific serum IgG geometric mean concentrations (GMCs), and the proportion of children with concentrations ≥ 0.35μg/mL (considered the serological correlate of protection against IPD), were calculated for each time point [18]."

Reviewer 2 Report

This was an interesting study looking at the possibility of using a dose of PPSV in the first year of life. Overall I think that a bit more information needs to be given to readers about “when” or “in what circumstances” a PPSV dose should be given in the first year. Because the dogma in most locations is that it shouldn’t be given prior to 2 years, we need more info about why it should be given to young infants – like why would immunological response be different in some areas compared to high income countries?

I would update reference 1 with latest info (note that findings for PNG are also available): McAllister, D. A.; Liu, L.; Shi, T.; Chu, Y.; Reed, C.; Burrows, J.; Adeloye, D.; Rudan, I.; Black, R. E.; Campbell, H.; Nair, H. Global, regional, and national estimates of pneumonia morbidity and mortality in children younger than 5 years between 2000 and 2015: a systematic analysis. Lancet Glob. Heal. 2018, 7, e47–e57, doi:10.1016/S2214-109X(18)30408-X.

Do you have a sample size calculation?

Have you done a per-protocol analysis? Anything different from the intention-to-treat analysis?

Table 1 – could you explain ppy? For instance a morbidity of 1.27 deaths per person – year doesn’t seem quite right (per 100 py? 1000 py?)

Author Response

This was an interesting study looking at the possibility of using a dose of PPSV in the first year of life. Overall I think that a bit more information needs to be given to readers about “when” or “in what circumstances” a PPSV dose should be given in the first year. Because the dogma in most locations is that it shouldn’t be given prior to 2 years, we need more info about why it should be given to young infants – like why would immunological response be different in some areas compared to high income countries?

Response:

The reviewer is correct that like other polysaccharide vaccines, in general the immunogenicity of PPV is limited in children under 2 years of age; however, as shown in previous studies and in this study, PPV is immunogenic (induces a steady increase in serotype-specific IgG responses) and associated with protection in young children in PNG. We cannot prove but only suggest that this is to some extent explained by the high level of pneumococcal exposure (contributing to immune maturation/adaptive immunity) in this population. We have added this explanation to the following paragraph in the discussion:

“While PPV does not induce immunological memory and its immunogenicity in young infants has limitations [21], PCVs are highly immunogenic in infants, induce immunological memory and antibodies with opsonophagocytic activity, and are effective in preventing IPD caused by vaccine serotypes in young children in low and high-risk countries [22-25]. Nevertheless, as shown here and by other studies, PPV is immunogenic and can induce high avidity, opsonophagocytic antibodies when given to PCV-primed infants under the age of 2 in high-risk settings and as young as 9 months of age in PNG [8, 19, 26-27], Importantly, early studies conducted in PNG before PCVs became available, have shown that PPV had a 45% protective effect against moderate/severe ALRI at the time of an epidemic [9]; a 50-59% efficacy against ALRI mortality [10]; and reduced overall mortality [28], suggesting these immune responses are protective. We suggest that the high level of pneumococcal exposure can be one of the explanations for the immunogenicity of PPV in young infants in high-risk settings such as in PNG”.

We suggest that administration of one dose of PPV before the age of 1 year could improve protection in infants living in high-risk settings and at high risk of severe pneumococcal infections caused by a broad range of serotypes.   

We believe that this is addressed in the concluding sentence of the manuscript:

“In summary, this study indicates that until PCVs with broader serotype coverage or serotype-independent vaccines become available, priming with currently available PCVs in early infancy followed by a PPV booster in late infancy is likely to be the best available strategy to protect young children in high-risk settings against severe pneumococcal infections caused by a broad range of pneumococcal serotypes.”

I would update reference 1 with latest info (note that findings for PNG are also available): McAllister, D. A.; Liu, L.; Shi, T.; Chu, Y.; Reed, C.; Burrows, J.; Adeloye, D.; Rudan, I.; Black, R. E.; Campbell, H.; Nair, H. Global, regional, and national estimates of pneumonia morbidity and mortality in children younger than 5 years between 2000 and 2015: a systematic analysis. Lancet Glob. Heal. 2018, 7, e47–e57, doi:10.1016/S2214-109X(18)30408-X.

Response: We thank the reviewer for this suggestion; however, while the suggested reference provides more recent estimates on the burden of pneumonia in children under 5 years of age in developing countries, it does not provide estimates for the burden of pneumococcal pneumonia (disease due to Streptococcus pneumoniae) as provided in reference 1. We have therefore kept the original reference.

Do you have a sample size calculation?

Response: The sample size calculation for this study has been described in the study protocol that was published in Pneumonia (Lehmann, D et al. Rationale and methods of a randomized controlled trial of immunogenicity, safety and impact on carriage of pneumococcal conjugate and polysaccharide vaccines in infants in Papua New Guinea Pneumonia 2017, 9 (20)).

We have added (Line 100) that sample size calculations are presented in the published study protocol.

For the reviewer’s interest: sample size calculations were based on an anticipated 80% follow-up rate with evaluable samples to the final visit based on results from our 7vPCV trial (87% at age 18 months). Based on the assumption that with respect to the primary outcome 90% of children would achieve serotype-specific antibody concentrations 0.35 µg/ml at 4 months of age, a sample size of 100 children (receiving 10vPCV or 13vPCV) was calculated to provide a 95% confidence that the true proportion is within 6% of this level of 90%. In view of the higher loss to follow-up than anticipated during the first year of the study, the total number of children to be enrolled was increased from 200 to 260.

Have you done a per-protocol analysis? Anything different from the intention-to-treat analysis?

Response: No, in line with the study protocol the analysis was by intention to treat and we did not perform a per protocol analysis.

Table 1 – could you explain ppy? For instance a morbidity of 1.27 deaths per person – year doesn’t seem quite right (per 100 py? 1000 py?)

Response: Table 1 provides Incidence rates of any morbidity, hospitalization, and any or moderate/severe acute lower respiratory tract infections (ALRI) between 9 and 24 months of age, expressed as per person year (ppy). This is not a mortality rate. That the rate is > 1 means that on average children had more than one episode of recorded illness per year.